# Agronomic Evaluation of Recycled Polyurethane Foam-Based Growing Media for Green Roofs

Patrice Cannavo [1,*], Mathieu Artous [1], Olivier Lemmel [2], Hervé Buord [3], Laure Vidal-Beaudet [1] and René Guénon [1]

[1]   Institut Agro, EPHOR, 49000 Angers, France
[2]   Brangeon Recyclage, 49300 Cholet, France
[3]   Jaulin Paysages, Chemin des Gruellières, 44470 Carquefou, France
*   Correspondence: patrice.cannavo@agrocampus-ouest.fr

**Abstract:** Green roofs are very popular and their individual surface area is constantly growing. Considering that the organo-mineral materials used in planting growing media (GM) are often non-renewable resources, the search for alternative materials from waste recycling is a challenge. Among these, recycled polyurethane (PU) foams are light and porous. The objective of this study was to evaluate the potential agronomic valorisation of PU foams for extensive green roofs. Three GM based on compost, PU foam and topsoil were developed and tested in situ for 18 months along with four plant species in containers containing 15 cm of GM. The agronomic properties of the GM and their contaminant contents were evaluated, as well as the plant aerial and root biomasses and trace element levels. The main results of this work are that GM are suitable for plant growth. Compost ratio effect resulted in a lower pH and higher exchangeable cations in GM1, whereas topsoil proportion effect mainly decreased macroporosity and increased nutrient contents. Furthermore, due to the high trace element load in the compost, hyperaccumulator plants such as *Hypericum calycinum* and *Stipa tenuissima* should be preferred. Ecotoxicological analyses will be carried out to validate the absence of risk of PU foam contaminants being released in the environment before proposing these types of GM to green roof developers.

**Keywords:** compost; organic matter; topsoil; plant biomass; trace elements; polycyclic aromatic carbon; decabromodiphenyl ether; lixiviation

## 1. Introduction

The world population is expected to reach 9.1 billion by 2050, 34% more than in 2017 [1]. This increase translates into growing urbanisation, and megacities are a major issue (e.g., pollution, urban heat islands) [2,3]. Several policies and regulatory schemes have been implemented in recent years with strong ambitions in terms of carbon storage and greenhouse gas (GHG) emission reduction. The European Commission adopted a series of proposals in 2019 to adapt EU climate, energy, transport and taxation policies with a view to reducing net GHG emissions by at least 55% by 2030 compared to 1990 levels. In France, the Energy and Climate Law (8 November 2019), and then the Climate and Resilience Law (2021), aim to achieve carbon neutrality by 2050, i.e., all carbon emissions in France should be offset by its absorption capacity. Nature-based solutions can ensure natural benefits such as soil protection, reduction in pollutant emissions, thermal regulation, improvement of air quality and maintenance of biodiversity [4]. The greening of urban spaces is a potential lever for action on the energy transition, especially as the societal demand asks for greening cities to adapt to climate change. Thus, urban spaces require fertile and sustainable planting growing media (GM) in terms of plant production capacity.

Green roofs are one specific type among vegetated urban ecosystems [5,6]. In addition to street trees or park areas, green (vegetated) roofs have been an important part of the

urban green infrastructure in recent years [6–8]. The numbers and surface areas of green roofs have been globally increasing in recent years. The annual growth in green roof covering is estimated to be 0.1 to 1 km$^2$ in several countries all over the world (Spain, Brazil, Canada, Korea, the UK or Japan), while it is estimated to reach 2 km$^2$ in France [9]. In Germany—a leading country in green roof construction—it is assumed that about 8 km$^2$ of green roof area are installed annually [10]. Green roofs are composed of extensive or intensive roof vegetation systems [8]. Extensive systems have shallow GM depths (2–20 cm) and are primarily vegetated with drought-tolerant sedums and mosses requiring limited maintenance, while intensive systems have deeper GM (>20 cm), are more diverse, not limited to specific plant types, and require regular maintenance and irrigation [8,11]. Green roof ecosystems have been characterized to provide a range of urban ecosystem services, e.g., microclimate regulation, air quality improvement, stormwater retention, habitat for flora and fauna, and aesthetic values [8]. However, the roofs of buildings are not designed to support heavy loads [12]. Therefore, the installation of planting GM can generate a mechanical stress, so that GM components that are both light and water retentive should be used. The load of an extensive green roof structure typically varies between 60 and 150 kg·m$^{-2}$ [13]. The materials used in GM are often non-renewable products such as mineral materials (pumice, zeolite, vermiculite, perlite), mixed with organic materials (peat, compost, biochar) [13]. Alternative materials to natural resources are currently being sought.

Invented in Germany in the 1930′s, polyurethane (PU) is a polymer-based material very different from other plastic materials. It is one of the most versatile materials in the world today. Its many uses range from flexible foam in upholstered furniture to rigid foam for insulation of walls, roofs and appliances, thermoplastic PU used in medical devices and footwear, as well as coatings, adhesives, sealants and elastomers used on floors and automotive interiors [14,15]. Among synthetic plastics, PUs rank sixth as the most common type of plastic used worldwide, and account for 6 to 7% (10 Mt/year) of the total plastics produced worldwide; European production represents 37% of worldwide production [16]. In France, annual production of PU was estimated to be 250 kt in 2011, and only 3 kt are recycled each year [17]. Years of research, study and testing have resulted in a number of recycling and recovery methods for PUs that can be economically and environmentally viable [18]. The four major categories are mechanical recycling, advanced chemical and thermo-chemical recycling, energy recovery and product recycling [19].

Ways of using these foams are being sought. Among them, the agronomic use of recycled PU in planting GM is being questioned. PUs have a very low density (20–30 kg·m$^{-3}$ on average) [20], so that they can be used to design light planting GM for green roofs, for example. They also have a high porosity (85% on average) [21], hence an interesting potential for GM aeration and water retention. These foams are already used as GM in hydroponic systems, as roots easily colonise them [22,23]. However, to our knowledge, the integration of PU foams into soils or substrates used as GM has never been studied.

Agronomic use could be a way of recycling PUs by developing crop supports for ornamental plants. However, the use of PU alone is not sufficient to mimic the properties of a soil. Horticultural soils and GM require a reserve of water and nutrients to meet the plant needs. Thus, a soil engineering approach has to be implemented to identify the relevant PU-based materials and mixing ratios in order to develop a GM for ornamental plants. All the materials in the GM design must comply with the various regulations applicable to these products.

The objective of this work is to evaluate the suitability of different PU-based GM for plant growth in extensive roof applications. A preliminary study carried out in a greenhouse identified relevant GM in terms of suitability for (1) plant growth and (2) bearing capacity for a green roof GM due to the lightness of the foams [24]. Following this, an 18-month in situ experiment was conducted to observe the development of four plant species on three different GM based on compost, topsoil and PU foams, respectively. The scientific

hypotheses were that the presence of foam would make the GM lighter, and the topsoil would allow for a better structuring of the mixture.

## 2. Materials and Methods

### 2.1. PU Foam Preparation

The foams came from the core of polyurethane mattresses, whose annual deposit is approximately 49 kt in France. The mattresses came from a deposit established by Ecomobilier, a non-profit eco-organisation created in December 2011 for collecting and recycling worn furniture in France. These mattresses could have been used for many purposes. Brangeon Fers Co., Cholet (France) received a batch of foams in 2019 that was representative of the global deposit. Two hundred $m^3$ (about 20 tons) were crushed on site, in two steps: (1) rough pre-crushing, and (2) shredding. For final shredding, the mesh size was 20 mm, but the particle size of certain foams after shredding was sometimes coarser because their soft nature made it difficult to reach homogeneous grinding.

### 2.2. Mix Preparation

In order to create and evaluate the agronomic value of PU foam-based GM, the foams were combined with 2 other materials—compost or topsoil (soil from the stripping of the first 0–30 cm horizon of an arable soil). A PU foam-compost mixture was initially developed and produced by Brangeon Fers Co., using standardised compost made from materials of agronomic interest derived from water treatment [25], in the following proportion: 2 volumes of screening refuse, 1 volume of green waste, and 1 volume of sewage sludge. The composting process included four stages: grinding, fermentation, maturation, and screening. The foams were sanitised before mixing to remove potentially present pathogens. The most commonly used conventional sanitisation method is steaming. The foams were incorporated into the composting process to undergo steam sanitization, and then the mixtures were matured for 9 weeks (forced aeration, maturation and screening of the material). Preliminary tests were carried out under controlled conditions in a greenhouse [24]. Different mixtures of compost with foam (100:0; 80:20; 60:40 and 40:60 by volume) and topsoil were installed in 3L containers and sown with ryegrass (*Lolium perenne*). In view of plant biomass production and the permissible loading values in extensive green roof systems (<180 kg·m$^{-2}$, [26]), 3 GM were selected for the trials. Their volume proportions are presented in Table 1. Two compost-foam mixtures M1 (60:40) and M2 (40:60) were mixed with topsoil (TS) collected at the Institut Agro Rennes-Angers (France). The physicochemical characteristics of these materials are presented in Table 2, and the contaminant contents in Table 3. The ecotoxicological analyses of M1 and M2 showed microbiological contaminants, trace metals, volatile organic compound (VOC), polychlorinated biphenyl (PCB), and polycyclic aromatic hydrocarbon (PAH) values all below the standard NF U44–551 thresholds or below the detection threshold (results not shown). The limits of detection were well below the thresholds of the standard.

**Table 1.** Characteristics of the growing media (GM). M = sanitised compost:foam mixes, TS = topsoil.

| Growing Media (GM) | Sanitised Compost: Foam Mixes (M) Vol. Ratio | Proportion of M in the GM (% vol.) | Proportion of TS in the GM (% vol.) | Load for 12 cm Thickness (kg·m$^{-2}$) [1] |
|---|---|---|---|---|
| GM1 | M1 60:40 | 80 | 20 | 129 |
| GM2 | M2 40:60 | 80 | 20 | 111 |
| GM3 | M2 40:60 | 60 | 40 | 179 |

[1] [24].

**Table 2.** Initial physico-chemical properties of the materials.

|  | **Compost** | **M1** | **M2** | **TS** | **Unit** |
|---|---|---|---|---|---|
| pH | 9.1 | 6.75 | 7.06 | 7.70 | - |
| Electrical conductivity | 9.1 | 2.15 | 2.6 | 0.10 | $dS \cdot cm^{-1}$ |
| Organic matter | 44.6 | 55.7 | 63.7 | 5.04 | $g \cdot 100 \, g^{-1}$ dw |
| Dry matter | 58.9 | 57.1 | 60.7 | 92.5 | $g \cdot 100 \, g^{-1}$ dw |
| Total nitrogen | 11.8 | 15.91 | 18.04 | 2.60 | $gN \cdot kg^{-1}$ dw |
| $NH_4^+$-N | 1.5 | 2.24 | 2.81 | <0.10 | $gN \cdot kg^{-1}$ dw |
| $NO_3^-$-N | 2300 | 495.2 | 338.9 | 52.0 | $mg \cdot kg^{-1}$ dw |
| B | 16.9 | 19.40 | 19.40 | 0.37 | $mg \cdot kg^{-1}$ dw |
| Co | 5.5 | 2.80 | 2.65 | 15 | $mg \cdot kg^{-1}$ dw |
| Fe | 11,594.7 | 9134.8 | 10,788.7 | 86.80 | $mg \cdot kg^{-1}$ dw |
| Mn | 192.9 | 248.9 | 203.4 | 21.20 | $mg \cdot kg^{-1}$ dw |
| Mo | 1.2 | 0.90 | 1.20 | 0.50 | $mg \cdot kg^{-1}$ dw |
| CaO | 121.7 | 24.05 | 20.33 | 7.15 | $g \cdot kg^{-1}$ dw |
| $K_2O$ | 16.2 | 14.93 | 10.60 | 0.52 | $g \cdot kg^{-1}$ dw |
| MgO | 7.75 | 4.66 | 3.65 | 0.39 | $g \cdot kg^{-1}$ dw |
| $Na_2O$ | 1.28 | 0.098 | 0.070 | 0.085 | $g \cdot kg^{-1}$ dw |
| CEC | 54.44 | 41.22 | 31.14 | 9.89 | $meq \cdot 100 \, g^{-1}$ dw |
| $P_2O_5$ Olsen | 0.71 | 1.53 | 1.06 | 0.16 | $g \cdot kg^{-1}$ dw |
| Bulk density | 0.44 | 0.20 | 0.16 | 1.28 | $g \cdot cm^{-3}$ |
| Total porosity | 79.94 | 88.71 | 90.53 | 49.72 | % vol |
| Macroporosity | 39.04 | 52.01 | 65.73 | 5.30 | % vol |
| Available water | 0.89 | 0.52 | 0.62 | 1.07 | $mm \cdot cm^{-1}$ |
| Water field capacity (−10 kPa) | 0.50 | 0.367 | 0.248 | 0.444 | % vol |

pH [27]; conductivity at 25 °C [28]; organic and mineral dry matters [29,30]; total nitrogen and ammonium nitrogen [31]; nitrate [32]; B boron soluble in boiling water [33]; trace metals [34]; Co and Mo [35]; Olsen phosphorus [34,35]; CEC (cobaltihexamine) [36]; exchangeable calcium, magnesium, potassium and sodium [34,37]; water retention curves [38].

**Table 3.** Contents in trace metals, volatile organic compounds (VOCs), sum of 7 polychlorinated biphenyls (PCBs), sum of 16 polycyclic aromatic hydrocarbons (PAHs) and decabromodiphenyl ethers (BDEs) of the compost, M1 and M2.

|  | **Compost** | **M1** | **M2** | **Unit** |
|---|---|---|---|---|
| Cd | 0.4 | 0.6 | 0.4 | $mg \cdot kg^{-1}$ dw |
| Cr | 24.8 | 20 | 21.8 | $mg \cdot kg^{-1}$ dw |
| Cu | 142 | 145.3 | 112.4 | $mg \cdot kg^{-1}$ dw |
| Hg | 0.3 | 0.097 | 0.189 | $mg \cdot kg^{-1}$ dw |
| Ni | 15.4 | 13.5 | 13.2 | $mg \cdot kg^{-1}$ dw |
| Pb | 23.8 | 23.2 | 22.4 | $mg \cdot kg^{-1}$ dw |
| Zn | 338 | 342.2 | 382.4 | $mg \cdot kg^{-1}$ dw |
| VOCs | <DL | <DL | 280.4 | $\mu g \cdot kg^{-1}$ dw |
| Sum of 7 PCBs | <DL | <DL | <DL | $\mu g \cdot kg^{-1}$ dw |
| Sum of 16 PAHs | <DL | 37 | 49 | $\mu g \cdot kg^{-1}$ dw |
| BDE 28 | – | 4.69 | 4.23 | $\mu g \cdot kg^{-1}$ dw |
| BDE 47 | – | 248.13 | 295.29 | $\mu g \cdot kg^{-1}$ dw |
| BDE 99 | – | 219.96 | 405.70 | $\mu g \cdot kg^{-1}$ dw |
| BDE 100 | – | 44.92 | 84.32 | $\mu g \cdot kg^{-1}$ dw |
| BDE 153 | – | 14.44 | 28.68 | $\mu g \cdot kg^{-1}$ dw |
| BDE 154 | – | 10.46 | 22.68 | $\mu g \cdot kg^{-1}$ dw |
| BDE 183 | – | 2.86 | 3.04 | $\mu g \cdot kg^{-1}$ dw |
| BDE 209 | – | 57.10 | 60.70 | $\mu g \cdot kg^{-1}$ dw |

Cd, Cr, Cu, Hg, Ni, Pb, Zn [38], VOCs [39], PCBs [40], PAHs [41], BDE [42].

## 2.3. Experimental Setup

The GM were prepared in a concrete mixer. They were placed in high-density polyethylene (HDPE) plastic containers (L = 111 cm; W = 71 cm; H = 61 cm), which allow for leachate recovery. In addition, HDPE is a plastic material that does not interact with any metallic pollutant. The containers were first filled with 40 cm of gravel (20–40 mm), then a geotextile was added, and then 15 cm of GM, in accordance with the GM of extensive roofs. The geotextile prevented the GM from leaching through the container and prevented the plant roots from colonising anywhere other than the GM. Four plant species commonly used in extensive green roofs were selected: ray-grass (*Lolium perenne*, LP), euphorbia (*Euphorbia cyparissias*, EU), St. John's wort (*Hypericum calycinum*, HY) and stipa (*Stipa tenuissima*, ST). Each condition was replicated 3 times and installed randomly in two rows of 12 containers each. Drip irrigation was used. Next to the rows of containers, a 30-cm deep trench was dug to place a bucket in front of each container so as to collect the leachates. This device was specific to each container. It consisted of a watertight connection at the bottom of the container, a one-metre long flexible pipe, and a 35-L HDPE bucket and its lid. This volume was chosen because it corresponded to the volume of water received by a tray (0.7881 m$^2$ surface area) during the heaviest rainfall in 24 h recorded over the 2010–2020 period at the MétéoFrance weather station of Beaucouzé located 300 m from the experimental platform.

The dynamic of rainfall, mean daily temperature and potential evapotranspiration (PET) is presented in Figure 1. Rainfall was evenly distributed across the study period, with a drier period in spring-summer (June–September 2020), except for the month of August when a major rainfall event of 46.6 mm occurred. Cumulative rainfall throughout the experiment (490 days) was 927.4 mm, while cumulative PET was 1042.3 mm. Average daily temperature varied between −2.1 and 28.4 °C.

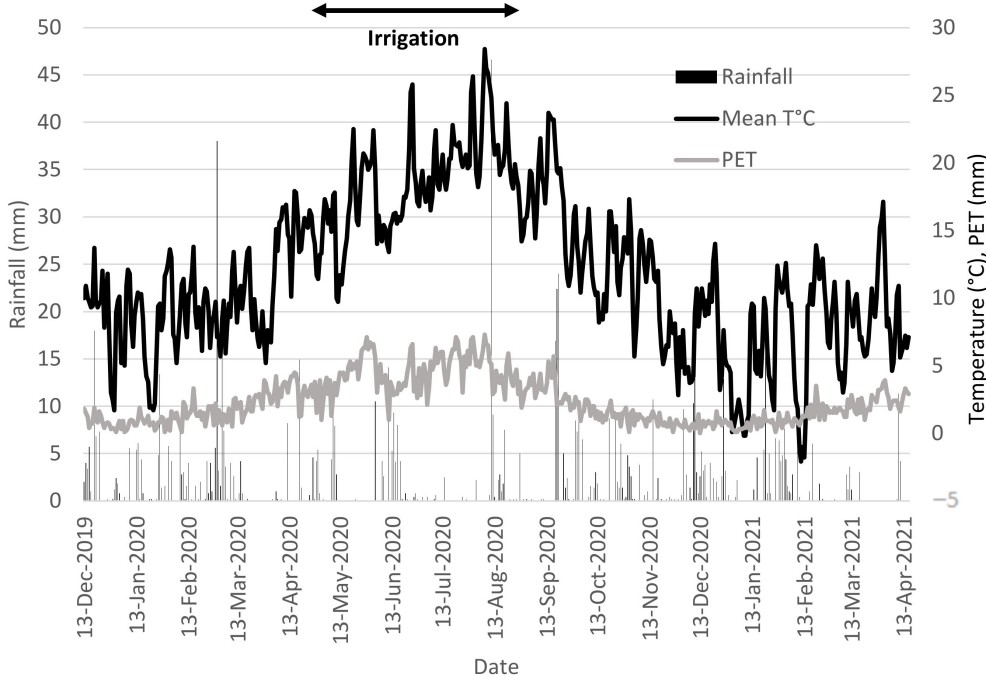

**Figure 1.** Weather conditions throughout the experiment. PET, potential evapotranspiration.

The plants were planted on 13 December 2019. The trial ended on 7 June 2021. Weeds were regularly removed manually.

## 2.4. Analyses

### 2.4.1. Leachates

Leachate collection is dependent on rainfall. Therefore, the sampling campaigns were more frequent in wet periods than in dry periods. From December 2019 to 17 March 2020,

leachates were collected every 1 or 2 weeks. From 17 March 2020, the COVID-19 lockdown in France disrupted the sampling protocol until 11 May 2020. However, this period of dry weather did not lead to drainage, but to the start of irrigation, so that no leachate was lost. Once the lockdown was relieved, leachates were collected when the collection buckets were full: 1% of the total volume of leachate was sampled per replicate. The samples were mixed to form composite samples grouping the 3 replicates and 2 consecutive collection times, and analysed throughout the experiment. The monitored parameters were the pH, electrical conductivity at 20 °C [43], $B^-$, $Ca^{2+}$, $Cu^{2+}$, $Mn^{2+}$, $Fe^{3+}$, $Mg^{2+}$, $PO_4^{3-}$, $K^+$, $SO_4^{2-}$, and $Zn^{2+}$ [34], $NO_3^-$ [32], $NH_4^+$ [43]. All these contents were compared to the standard of water potability set by [44].

### 2.4.2. Plant Analysis

At the end of the experiment, the aboveground biomass of each plant was harvested and weighed fresh and dry after drying at 60 °C in an oven for 48 h. The roots were washed with de-ionised water, dried at 60 °C in the oven for 48 h, and weighed. Aboveground and root biomass samples were ground and analysed for their N, P, K, Mg, Ca, Cu, Zn, Mn, B, Fe, Cr, Hg, Ni and Pb contents [45]. As the plants were ornamental plants—not intended for food use—we did not compare the levels of these elements to the admissible levels for humans.

### 2.4.3. Growing Medium Analysis

Before the plants were uprooted, three undisturbed cores (5 cm diameter, 6 cm height) were taken at 5 cm depth to estimate bulk density in each container. The soil water content was determined after oven-drying the samples at 105 °C for 48 h, and bulk density values were calculated from the dry soil mass and the cylinder volume [46]. Three other undisturbed cores (7 cm diameter, 5 cm height) were sampled per container at 5 cm depth. They were used to estimate the soil volumetric water content ($m^3$ $m^{-3}$) at saturation ($\theta_s$) (−0.1 kPa), at field capacity ($\theta_{fc}$) (−1 kPa), and at the temporary wilting point ($\theta_{wp}$) (−10 kPa) according to [47]. Available water (AW) for plants (mm of water $cm^{-1}$ of GM) (Equation (1)) and macroporosity (Pmacro) ($m^3$ $m^{-3}$) (Equation (2)) were calculated as follows:

$$AW = \theta_{fc} - \theta_{wp} \tag{1}$$

$$Pmacro = \theta_s - \theta_{fc} \tag{2}$$

After removal of the plants, one kilogram of composite GM sample was collected from each container across the entire GM thickness in triplicate and was air-dried for analyses as described in Table 2. Ecotoxicological analyses were also performed. These included trace metals As, Cd, Cr, Cu, Hg, Ni, Zn [38], VOCs [39], PCBs, PAHs [41,42] and 8 decabromodiphenyl ethers (BDEs) [48]. The pollutant levels were compared with contamination thresholds found in the scientific literature. Trace metal contents were compared with the limits set by the French organic amendment standard NFU 44-051 [49]. PCBs are not subject to a critical threshold within the NFU 44-051 standard, but they are subject to regulations for composts containing materials of agronomic interest resulting from sludge water treatment [25]. As soil PAH levels are not subject to regulation, there is no threshold for identifying PAH contamination in soils. However, the ubiquitous concentrations linked to natural sources are in the order of 0.1 to 1 mg·kg$^{-1}$ for the sum of 16 PAHs [50]. As regards BDEs, the European Union limits their maximum concentration [51] and limits the BDE concentration to 1 mg·kg$^{-1}$. Finally, we did not find any recommendation threshold for soil VOCs.

### 2.5. Statistical Analyses

We studied the effects of the compost-foam mixture and topsoil in a 2-factor analysis of variance (ANOVA) ($p < 0.05$) on the physico-chemical properties of the different mixtures and on the plant aboveground and root biomasses, using Statistica 13.0 software. Two

ANOVAs were performed, one to assess the effect of the PU foam proportion using GM1 and GM2, the other to assess the effect of the proportion of topsoil using GM2 and GM3. Among the mixtures, comparisons of means between plant biomasses were evaluated with Tukey's HSD (honestly significant difference) test at $p < 0.05$. Finally, a principal component analysis (PCA) was performed with FactoMineR package [52].

## 3. Results

### 3.1. Agronomic Properties of the Growing Media

The properties of the GM are presented in Table 4, and toxicological analyses in Supplementary Table S1. Significant differences between GM1 and GM2 were found in relation to the GM properties (ANOVA of the compost:foam ratio effect, GM factor, Supplementary Table S2). The most significant differences ($p < 0.01$) were observed for the pH (lower in GM1 than in GM2), the exchangeable nutrients MgO and $P_2O_5$ (higher in GM1 than in GM2), and the trace metal elements Cr (lower in GM1) and Cu (higher in GM1). Considering the topsoil proportion effect (i.e., GM2 vs. GM3), the GM factor impacted all GM properties significantly ($p < 0.01$) except for the pH and the water reserve (ANOVA, Supplementary Table S3). Strong differences ($p < 0.001$) were observed for the contaminant contents (Cd, Cu, Hg and almost all BDEs; lower in GM3 than in GM2), physical properties (bulk density and macroporosity; lower in GM2 than in GM3), and nutrients (OM, all exchangeable cations, CEC, $NH_4^+$, B and Mn; higher in GM2 than in GM3).

The pH values were quite similar for all conditions (7.6–7.8), a range in accordance with recommendations. Similarly, electrical conductivity was below 0.5 mS·cm$^{-1}$ in all treatments. The OM contents varied between 7.6% (GM3) and 9.7% (GM1, the GM containing the greatest proportion of compost) (Table 1), but remained within the recommended range. The CEC was generally below 40 meq 100 g$^{-1}$, below the recommended range, and varied between 15.3 (GM3) and 30.8 (GM1) meq 100 g$^{-1}$. As for exchangeable cations, all GM displayed values above recommendations, except for CaO which was below 8380 mg·kg$^{-1}$ in GM3. Concerning physical properties, all GM had a bulk density well below 1.2 g·cm$^{-3}$. This was particularly true in GM1 and GM2 in which the share of topsoil was 20% per volume, versus 40% in GM3. The water reserve was lower than the expected recommendation of 1.5 mm·cm$^{-1}$ whatever the GM. Conversely, macroporosity was highest in GM3, well above the recommended threshold, whereas GM1 and GM2 had values below the recommended 20% per volume. Finally, we calculated GM loading at water holding capacity from GM bulk density and available water. Whatever the GM, this load was less than 150 kg·m$^{-2}$, in accordance with the mechanical rules for a green roof (Table 4).

Toxicological analyses showed that no pollutant exceeded the recommendations, except for As whose levels were slightly higher than 18 mg·kg$^{-1}$, particularly in GM3. GM2 concentrated most BDEs and PAHs. No PCB was detected (Table S1).

**Table 4.** Main physico-chemical properties of the growing media ($n = 3$).

| | GM1-EU Mean | SD | GM1-HY Mean | SD | GM1-ST Mean | SD | GM1-LP Mean | SD | GM2-EU Mean | SD | GM2-HY Mean | SD | GM2-ST Mean | SD | GM2-LP Mean | SD | GM3-EU Mean | SD | GM3-HY Mean | SD | GM3-ST Mean | SD | GM3-LP Mean | SD | Recommendation |
|---|---|---|---|---|---|---|---|---|---|---|---|---|---|---|---|---|---|---|---|---|---|---|---|---|---|
| pH water | 7.6 | 0 | 7.7 | 0 | 7.6 | 0 | 7.7 | 0.1 | 7.8 | 0.1 | 7.8 | 0 | 7.8 | 0 | 7.7 | 0 | 7.9 | 0.1 | 7.8 | 0 | 7.8 | 0.1 | 7.7 | 0 | 6.5–7.5 [53] |
| Electrical conductivity ($mS \cdot cm^{-1}$) | 0.2 | 0 | 0.2 | 0 | 0.2 | 0 | 0.2 | 0 | 0.2 | 0 | 0.2 | 0 | 0.2 | 0 | 0.2 | 0 | 0.1 | 0 | 0.1 | 0 | 0.1 | 0 | 0.1 | 0 | <0.5 [36] |
| OM (%) | 9.7 | 0.5 | 7.5 | 0.5 | 10.3 | 1.1 | 9.5 | 1 | 7.7 | 0.4 | 8.7 | 0.9 | 9 | 0.2 | 9.6 | 0.9 | 7.6 | 0.4 | 7 | 0.4 | 7.1 | 0.5 | 6.3 | 0.1 | 4–10 [53] |
| $NO_3^{-}$-N ($mg \cdot kg^{-1}$) | 9.2 | 1.9 | 23.6 | 10.2 | 29.9 | 12.1 | 23.6 | 16.2 | 5.2 | 2.3 | 21.5 | 2.9 | 23.5 | 8.4 | 17.5 | 5.3 | 6.6 | 2.6 | 11.1 | 7.2 | 12.7 | 5.3 | 9.1 | 1.9 | - |
| $NH_4^{+}$-N ($mg \cdot kg^{-1}$) | 6.5 | 1.3 | 7.7 | 1.1 | 4.3 | 0.2 | 6.2 | 0.6 | 8.3 | 1.1 | 7.1 | 0.5 | 5.3 | 1 | 7 | 0.4 | 2.3 | 1.3 | 4.4 | 0.9 | 3.6 | 1.4 | 3.8 | 0.9 | - |
| CaO ($mg \cdot kg^{-1}$) | 9182.3 | 888.4 | 10,356 | 875.8 | 9161 | 492.2 | 9299.7 | 166.2 | 9394.3 | 617.1 | 9039.3 | 125.9 | 8298 | 370.2 | 10,534 | 3276.7 | 6478.7 | 586.1 | 6061 | 361.2 | 5479.3 | 429.7 | 4789.7 | 157 | 8379 [36] |
| $K_2O$ ($mg \cdot kg^{-1}$) | 444.3 | 50.9 | 346 | 40 | 326.3 | 7.2 | 315 | 56 | 448.3 | 40.3 | 339.7 | 40.8 | 287.3 | 22.2 | 264.7 | 25.5 | 314 | 30.3 | 230.3 | 25.3 | 229 | 13.5 | 230.3 | 8.3 | >0.18 [54] |
| MgO ($mg \cdot kg^{-1}$) | 702.3 | 61.9 | 734.7 | 53.5 | 743.3 | 28.6 | 585.3 | 37.7 | 637 | 28.9 | 563 | 35.7 | 526.3 | 9.9 | 540.7 | 35.1 | 457.3 | 50.7 | 393 | 8 | 381 | 23.8 | 363.7 | 14.5 | 119 [36] |
| $Na_2O$ ($mg \cdot kg^{-1}$) | 105.3 | 15 | 102 | 7.6 | 96.3 | 4.6 | 91 | 3.7 | 102 | 5.1 | 94.3 | 7.8 | 89 | 8.2 | 86.7 | 2.5 | 69.3 | 4.6 | 66.7 | 2.5 | 59 | 3.7 | 57.7 | 1.1 | <355 [36] |
| CEC ($meq\ 100\ g^{-1}$) | 27.2 | 4.9 | 30.4 | 0.8 | 30.8 | 0.9 | 26.5 | 1.4 | 25.3 | 2.8 | 26.5 | 1.9 | 23.3 | 3.2 | 23.5 | 0.9 | 18.5 | 0.9 | 16.4 | 0.6 | 16.5 | 2.9 | 15.3 | 1 | >40 [53] |
| $P_2O_5$ ($mg \cdot kg^{-1}$) | 486.3 | 24.9 | 476.7 | 31.2 | 517 | 61 | 478.3 | 16 | 411 | 50.4 | 415.3 | 11.5 | 430.3 | 12.7 | 406 | 11 | 275 | 8.3 | 254 | 3.1 | 254.3 | 8.2 | 245.7 | 1.8 | >120 [53] |
| B ($mg \cdot kg^{-1}$) | 0.7 | 0 | 0.7 | 0 | 0.8 | 0.1 | 0.6 | 0 | 0.8 | 0.1 | 0.7 | 0 | 0.7 | 0.1 | 0.7 | 0.1 | 0.5 | 0.1 | 0.5 | 0 | 0.4 | 0 | 0.5 | 0 | 0.4 [36] |
| Fe ($mg \cdot kg^{-1}$) | 116 | 5.1 | 124.9 | 9.2 | 120.7 | 4.2 | 118.3 | 7.8 | 125.4 | 5.3 | 112.5 | 20.6 | 142.7 | 4.5 | 111.3 | 41.1 | 136.8 | 18.2 | 146 | 16.4 | 157.3 | 12.9 | 184.8 | 18.3 | 9−300 [55] |
| Mn ($mg \cdot kg^{-1}$) | 19 | 2.7 | 19.3 | 2.2 | 17.5 | 1.5 | 19 | 1.2 | 25.8 | 2.5 | 21.3 | 1.6 | 21 | 2.2 | 20.9 | 2.8 | 34.9 | 2.1 | 29.9 | 2.7 | 30.5 | 0.9 | 33.3 | 3.9 | 9 [36] |
| Mo ($mg \cdot kg^{-1}$) | 0.1 | 0 | 0.1 | 0 | 0.1 | 0 | 0.1 | 0 | 0.1 | 0 | 0.1 | 0 | 0.1 | 0 | 0.1 | 0 | 0.1 | 0 | 0.1 | 0 | 0.1 | 0 | 0.1 | 0 | - |
| Co ($mg \cdot kg^{-1}$) | 0.5 | 0.1 | 0.4 | 0.1 | 0.3 | 0.1 | 0.5 | 0.1 | 0.7 | 0.1 | 0.4 | 0.2 | 0.3 | 0.1 | 0.5 | 0.2 | 0.5 | 0.1 | 0.4 | 0.1 | 0.4 | 0 | 0.4 | 0 | - |
| Bulk density ($g \cdot cm^{-3}$) | 0.53 | 0.04 | 0.58 | 0.02 | 0.57 | 0.06 | 0.5 | 0.01 | 0.5 | 0.06 | 0.56 | 0.04 | 0.55 | 0.03 | 0.49 | 0.03 | 0.81 | 0.04 | 0.82 | 0.07 | 0.88 | 0 | 0.82 | 0.06 | <1.2 [53] |
| Available water ($mm \cdot cm^{-1}$) | 1.06 | 0.28 | 1.12 | 0.32 | 1.22 | 0.24 | 0.68 | 0.03 | 1.16 | 0.2 | 1.13 | 0.15 | 0.99 | 0.22 | 0.92 | 0.07 | 0.85 | 0.17 | 0.88 | 0.16 | 0.94 | 0.11 | 1.07 | 0.25 | >1.5 [53] |
| Macroporosity (% vol) | 17.03 | 3.92 | 14.22 | 11.25 | 6.27 | 1.17 | 19.73 | 7.27 | 9.89 | 1.92 | 15.25 | 6.08 | 16.78 | 9.42 | 13.84 | 7 | 31.36 | 4.26 | 31.71 | 1.85 | 31.85 | 1.54 | 28.45 | 4.55 | >20 [53] |
| GM loading at water retention capacity ($kg \cdot m^{-2}$) | 100.1 | 11.4 | 108.5 | 9 | 108.6 | 13.6 | 88.5 | 2.2 | 97 | 12.1 | 105.6 | 8.6 | 101.5 | 8.1 | 90.8 | 6.3 | 137.7 | 8.9 | 140.4 | 14.2 | 150.3 | 2.4 | 143.4 | 13.2 | <180 French regulation [26] <150 [13] |

### 3.2. Plant Production

The aboveground and root biomasses of the plants are presented in Figure 2. The GM factor did not altogether affect plant biomass or elemental contents (Tables S4–S7). Only the topsoil ratio significantly affected the Cu ($p < 0.05$), Mn ($p < 0.05$) and Pb ($p < 0.01$) contents in the roots (Table S7). By contrast, the plant factor had a significant effect on almost all measured parameters. For aerial biomass, the compost:foam ratio and the topsoil ratio affected all parameters but a few trace metals—Hg (Table S4) and Cd, Cr, Hg (Table S5), respectively. For the root biomass, the compost:foam ratio and the topsoil ratio also affected all parameters except a few trace metals—Cd, Cr, Hg, Ni (Tables S6 and S7). The plant factor had a significant effect on very few GM properties, i.e., the two exchangeable cations MgO ($p < 0.05$) and K$_2$O ($p < 0.001$), whatever the proportions of the materials (Tables S2 and S3).

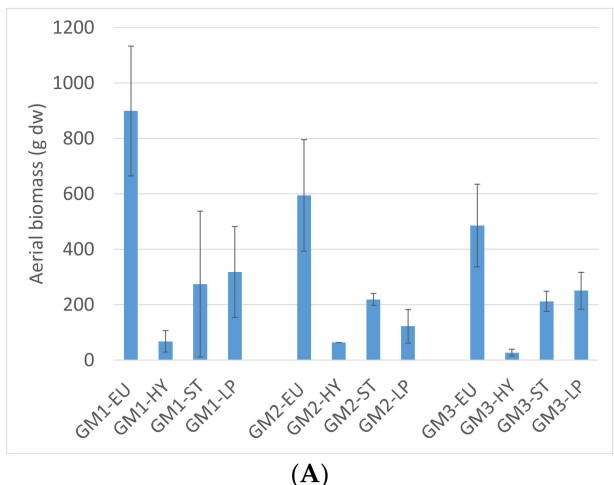 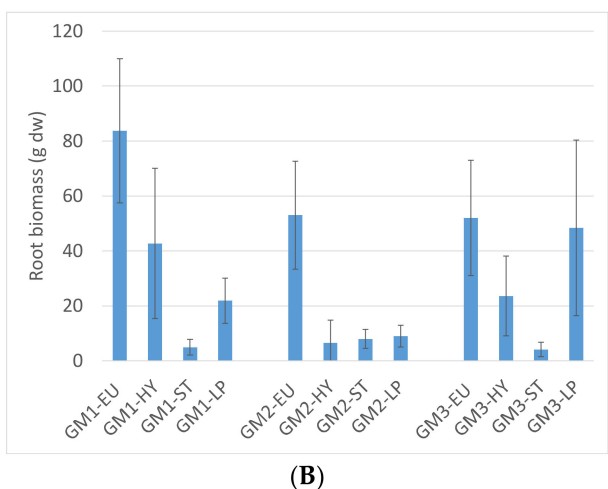

(**A**)        (**B**)

**Figure 2.** Aerial (**A**) and root (**B**) biomasses of *Euphorbia* (EU), *Hypericum* (HY), *Stipa* (ST) and *Lolium perenne* (LP) grown in the 3 GM.

Regardless of the conditions, all four plant species grew successfully in all three GM. For *Euphorbia* (EU) and *Hypericum* (HY), the variability of the measurements was too high to detect any effect of the GM on plant growth. *Lolium perenne* (LP) grew more in GM1 and GM3 than in GM2 ($p < 0.05$). *Stipa* (ST) did not produce different biomasses depending on the GM. The macro-element and trace metal contents of aerial biomass and root biomass are presented in Tables S8 and S9, respectively. The plant species showed a contrasting, element-dependent absorption capacity. HY and LP concentrated elements in higher quantities in their aerial parts than the other two species did, particularly the metals Fe, Zn, Cu, Mn, Cr and Zn ($p < 0.001$). The Fe content of LP was very high; it varied between 1168 and 4700 mg·kg$^{-1}$ in its aerial parts depending on the GM, and between 2425 and 5333 mg·kg$^{-1}$ in its roots. ST had the lowest elemental content ($p < 0.01$). As for the root biomass, concentrations—particularly metal concentrations—were generally highest in LP. The highest metal contents (Fe, Zn, Cu, Mn, Cd, Cr and Pb) were measured when it was grown in GM3, both in its aerial and root biomasses.

An ANOVA depicted very few GM×plant interactions. Depending on the topsoil ratio, interactions were observed in the root biomass, with a significant impact on Cu, Mn and B ($p < 0.05$) and Ca and Pb ($p < 0.01$) (Table S7).

### 3.3. Leachate Quality

Cumulative data for water fluxes, i.e., rainfall (R), potential evapotranspiration (PET), and drainage (D), between two leachate sampling dates are presented in Table 5. Drainage was fairly comparable between GM, with an average value of 323 mm representing 35% of rainfall. The amount of water brought by irrigation between May and September 2020 was not measured; this irrigation water did not lead to drainage.

**Table 5.** Cumulative values of rainfall (R), drainage and potential evapotranspiration (PET) between each leachate collection date since 13 December 2019. SD, standard deviation (*n* = 3).

| Collection Date | R | PET | Drainage | | | | | |
|---|---|---|---|---|---|---|---|---|
| | | | GM1 | | GM2 | | GM3 | |
| | | | Mean | SD | Mean | SD | Mean | SD |
| 11-Feb-20 | 153.3 | 52.5 | 60.5 | 2.5 | 52.9 | 3.6 | 57.2 | 2.8 |
| 25-Feb-20 | 22.4 | 21.1 | 44.4 | 2.5 | 42.5 | 2.4 | 45.7 | 1.6 |
| 12-May-20 | 205.6 | 215.6 | 37.0 | 0 | 37.0 | 0 | 37.0 | 0 |
| 1-Sep-20 | 155.4 | 523.7 | 37.0 | 0 | 37.0 | 0 | 37.0 | 0 |
| 18-Nov-20 | 189.8 | 160.8 | 37.0 | 0 | 37.0 | 0 | 37.0 | 0 |
| 13-Jan-21 | 107.9 | 34.8 | 37.0 | 0 | 37.0 | 0 | 37.0 | 0 |
| 17-Feb-21 | 93.0 | 33.8 | 37.0 | 0 | 37.0 | 0 | 37.0 | 0 |
| 16-Apr-21 | 40.9 | 125.5 | 37.0 | 0 | 37.0 | 0 | 37.0 | 0 |
| Total | 927.4 | 1042.3 | 326.7 | | 317.2 | | 324.7 | |

The characteristics of the leaching water are presented in Figure 3. Whatever the treatment, the pH gradually increased over the course of the experiment, by about 1 pH unit. The values remained in the agronomic range favourable for the growth of ornamental plants. Electrical conductivity decreased over time in all modalities. It was initially higher in GM1—around 1 mS·cm$^{-1}$—versus around 0.5 mS·cm$^{-1}$ in the other GM. In all cases, EC was below the recommended maximum of 2.5 mS·cm$^{-1}$. $NO_3^-$ concentrations were above the drinking water standard in GM1 during the first 3 months of monitoring. Regardless of the GM, the concentrations progressively decreased, and they were near 0 from 1 September 2020 onwards. The $NH_4^+$ concentrations evolved differently. Initially low in all GM, they gradually increased from spring onwards to peak in September. A further increase started in spring of the following year (2021). The peak concentrations reached in 2020 were particularly high in GM1 and GM2, above the recommended limit. The $Fe^{3+}$ and $Mn^{2+}$ concentrations followed the same pattern, i.e., a high concentration far above the recommendation at the start of the measurements, followed by a gradual decrease and then stabilisation around the recommended threshold after 18 November 2020. The starting concentrations were highest in GM1. The other elements ($PO_4^{3-}$, $SO_4^{2-}$, $K^+$, $Mg^{2+}$, $Ca^{2+}$, $B^-$, $Cu^{2+}$, $Zn^{2+}$, $B^{3+}$) showed the same dynamics as $Fe^{3+}$ and $Mn^{2+}$, i.e., a rapid decrease, but down to levels below the recommended maximums (Supplementary Tables S10–S17).

*3.4. Correlation between Plant Biomass and the Agronomic Properties of the GM*

The 2-dimensional projection of the PCA of plants and GM parameters measured after 18 months of cultivation explained 55% of the variance (Figure 4). The first axis was driven by GM nutrients and physical properties, and the second axis was mainly driven by biomass (Figure 4A). The representation of the individuals clearly distinguished the effect of the GM, with a separation mainly evidenced by the first axis (Figure 4B). For GM2 and GM3, the associated individuals were particularly clustered, not dispersed by the plants. For GM1, differences among individuals were greater.

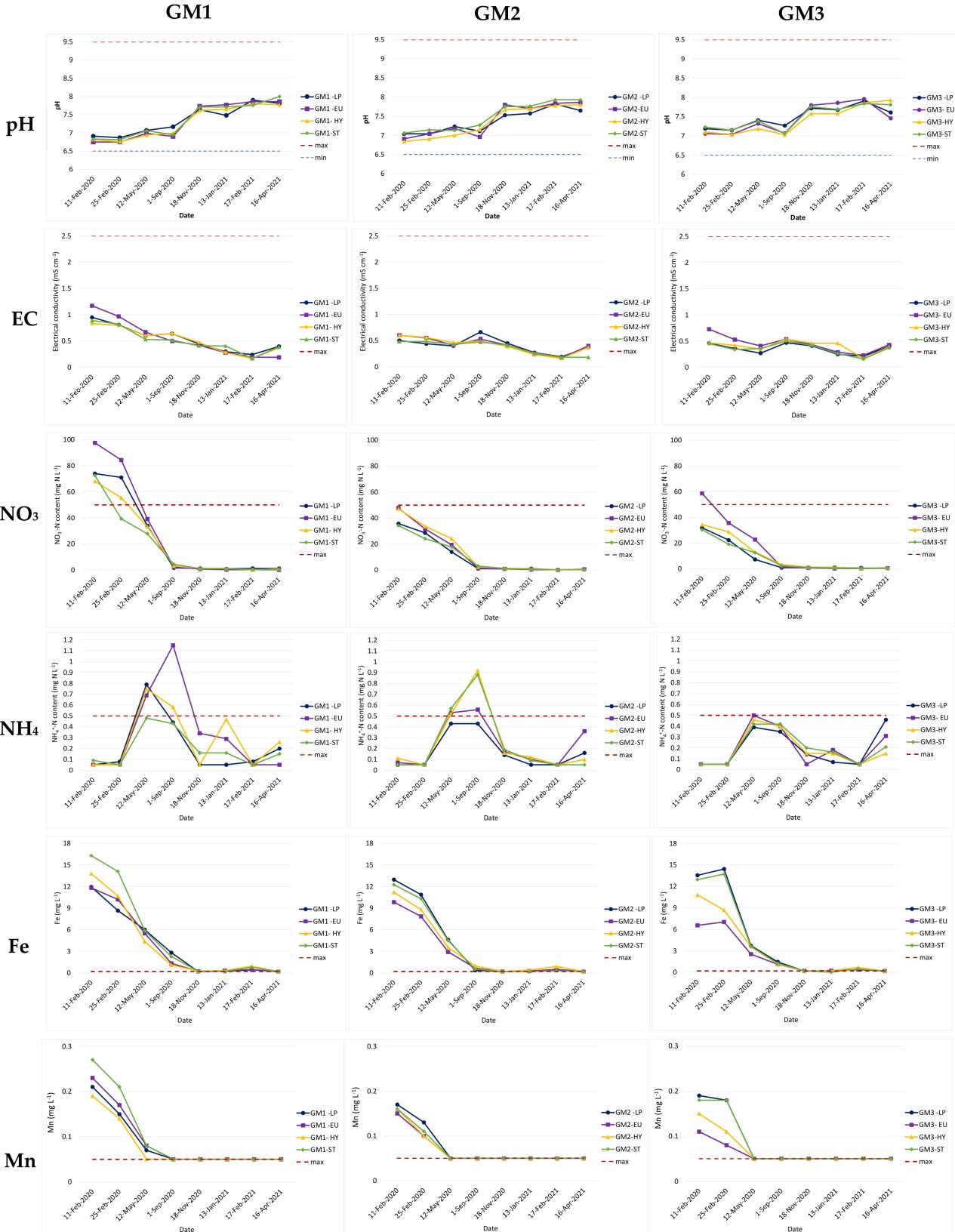

**Figure 3.** Dynamic of the pH, electrical conductivity (EC), and the $NO_3^--N$, $NH_4^+-N$, iron (Fe) and manganese (Mn) contents in the leachates. Maximum and/or minimum threshold contents correspond to the drinkable water standard [44].

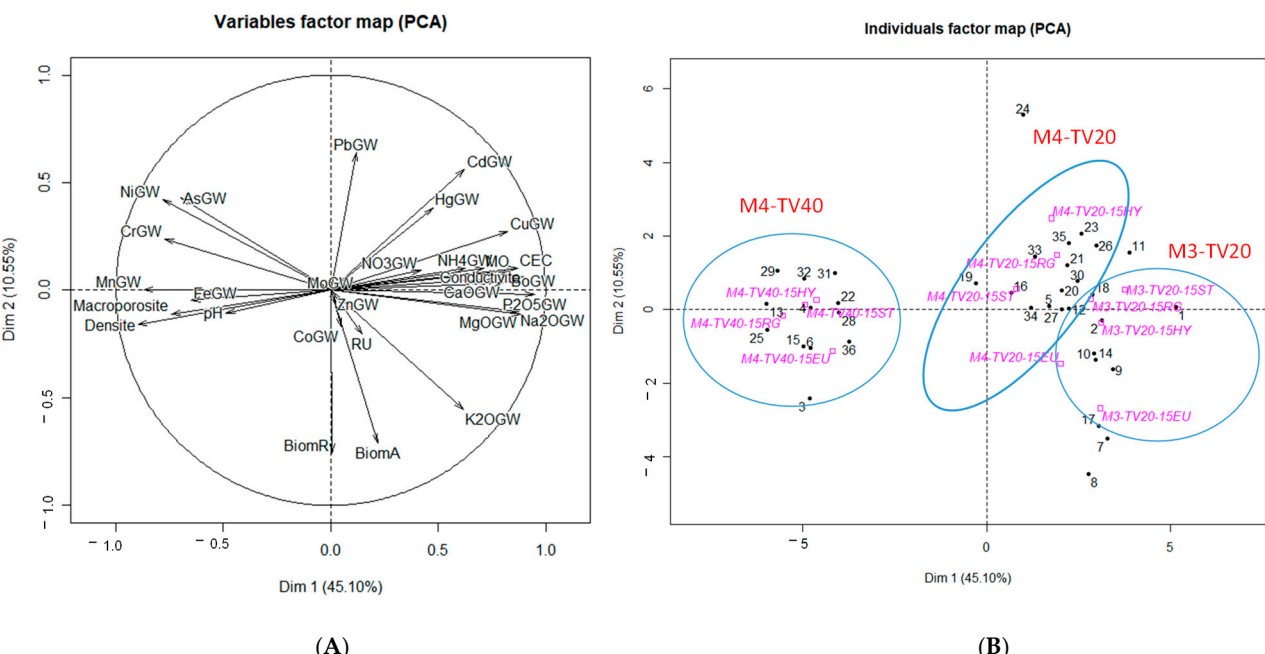

**Figure 4.** Correlation circle (**A**) and factorial map (**B**) as constructed from the principal component analysis of treatments and GM parameters measured at the end of the experiment. Individuals 1–12, 13–24, and 25–36 correspond to GM1, GM2 and GM3, respectively.

## 4. Discussion

### 4.1. Agro-Environmental Quality of the Growing Media

The 3 GM were mainly distinguished by their contrasting nutrient loads, as shown by the PCA. This nutrient load was particularly high in the GM with a low proportion of TS (GM1 and GM2). The difference between GM1 and GM2 was that GM1 contained more compost than GM2. The agronomic analyses showed that the two GM had acceptable physico-chemical values in relation to the standard. The high levels of exchangeable cations and trace elements were attributed to the compost, which was an important reservoir for these elements (Table 2). As far as $NO_3^-$-N is concerned, the high initial levels certainly resulted from OM mineralisation following the conditioning of the sanitised mixtures until their use in the experiment. The addition of foams resulted in a decrease in these contents.

Foams were the main source of PAH and BDE pollutants (Table 3), and their levels were highest in GM2 which contained the highest proportion of foam. This is explained by the nature of the foams which contained BDE-based flame retardants. Nevertheless, these pollutant levels remained below the standards. Since 2012, pentaBDE has been included in the list of chemicals and articles banned by EU Regulation No. 649/2012 of 4 July 2012. In France, Decree No 2004–1227 of 17 November 2004 prohibits the marketing of manufactured products or parts of manufactured products acting as flame retardants containing BDEs in concentrations exceeding 0.1% of the total mass. The recycled PU foams used in this project may have been produced before or after this decree was passed, so that BDE contents may vary from one waste to another. We assumed that the batch of waste received in 2019 (approximately 20 tons) was representative of the overall deposit. The foams also had an impact on the physical properties of the GM, with a decrease in bulk density and a decrease in macroporosity. On the other hand, the presence of a larger quantity of topsoil (40% vol., GM3) induced a structuring of the GM towards higher macroporosity.

The presence of trace metals generally follows from compost addition. This was particularly true for Cr, Cu, Hg, Pb and Ni (Table 3), while the initial Zn levels were higher in M1 and M2 than in the foam-free compost (Table 3), suggesting a Zn reserve within the foams. Nevertheless, the levels were all below the standards at the end of the study.

The analysis of the leaching water revealed a significant release of compounds during the first 6 months of the experiment, likely to be transferred to the environment and drinking water. This was particularly true for $NO_3^-$, Fe and Mn whose levels in the GM were highest. Concerning the leached nitrogenous forms, the dynamic of $NH_4^+$-N release was representative of the dynamic of OM mineralisation, with concentration peaks occurring during the spring-summer period, when temperature and humidity were favourable to this process [56]. Thus, an environmental risk of nutrient release into leaching water is possible if the composts have a high nutrient load at the time of installation on the green roof. A few studies have addressed the analysis of GM leachates. Nutrients and trace metals were their main concern [57–59]. Nevertheless, comparing our results is difficult because the materials used in these GM, their thickness and climate conditions differed from those of the present study. The time course of leached elements was investigated in [57]. However, the authors did not observe a common trend for all monitored elements, or a continuous decrease in elements as observed in our study. They suspected a combined effect of the quantities of water entering the system and the elements available in the GM at a given time.

We changed the collection method during this study because of the COVID-19 pandemic and the subsequent first lockdown. During the winter period and before the COVID-19 crisis (13 December 2019 to 25 February 2020), an average drainage value was determined for all species. This was not a problem because the plants did not transpire during that period, so there was only a GM effect. During the pandemic, the samples were taken when the buckets were full. The four plant species probably had different crop coefficients, and the amount of transpired water was not the same, especially in summer. Therefore, some buckets may have been full earlier than the sampling date and leachates may have been lost. However, this does not modify the trend of gradually decreasing elemental concentrations over time.

### 4.2. Ability of the Plants to Grow in Polyurethane Foam-Based GM

Although green roofs may be installed for many reasons, stormwater mitigation is altogether a key driver of their adoption to reduce the risk of flooding in built-up catchments and damage to waterways [60,61]. Green roofs reduce stormwater runoff by retaining rainfall in growing media or substrates and releasing it into the atmosphere via evapotranspiration [62]. However, plant species need to tolerate periods of water deficit between rainy spells to survive [63]. To date, the most common plants used on green roofs worldwide are succulents, mainly Sedum species [64,65]. They tolerate drought by using little water and storing water in their leaves [66]. Therefore, appropriate plant selection is critical to maximise both stormwater mitigation and plant survival on green roofs [67]. Other plants than sedum, selected from habitats analogous to extensive green roofs are also suitable. For example, an analysis of species from shallow soil habitats in southern France identified 28 plant species potentially suitable for hot and dry Mediterranean green roofs [68]. We selected plant species adapted to the climate of the study site, according to the recommendations of landscaping companies. The biomass produced in this experiment cannot be interpreted by comparing the plant species with each other because they have different morphologies and growth rates. One of the major results is that all 4 plant species grew successfully in all 3 GM, and the GM had no effect on biomass production. Normal growth was expected from LP because it had been successfully grown in the selected GM in a previous greenhouse trial [24]. HY was the trickiest plant to grow. Some individuals died in GM1 (1 replicate out of 3) and especially in GM2 (2 replicates out of 3). However, irrigation was interrupted by mistake in August following a failure of the irrigation system. HY was probably more sensitive than the other species to water stress.

Another important result is the contrasting trace metal accumulation *per* plant, and the promoting effect of GM3 on metal accumulation by the plants. HY and LP concentrated the highest metal contents, especially Fe and Mn. Such species are well known to accumulate high metal quantities in both their aerial parts and roots. LP is commonly used in the

restoration of degraded or contaminated soils, it is robust and low sensitive to pollutants and water stress [69–71], while HY is known for high metal bioaccumulation [72]. GM3 contained more topsoil than GM1 and GM2. It also contained more Fe and Mn; this probably explains the high Fe and Mn levels in the plants. By contrast, Zn and Cu—present at high levels in HY and LP—were in lower contents in GM3 than in GM2. We suppose that arable soil improved the structuring capacity of the GM and favoured metal solubilisation and bioavailability. Furthermore, HY and LP rhizosphere activity may have favoured metal uptake.

*4.3. Implications for the Foam Recycling and Green Roof Industries*

This study investigates a new–agronomic–way of recycling used PU foams. To our knowledge, no other study has explored such a solution. Although composts have to comply with the NFU 44-095 standard [25], they can still have very high nutrient loads when used as/in GM. Therefore, it is important to allow the mixture to mature for a few weeks under in situ conditions and in the absence of plants, so that elements in excess can leach out.

The trial by [24] showed that the more TS was present in the GM, the more LP biomass was produced. We chose a maximum TS proportion of 40% *per* volume so as to have a GM weighing less than 180 kg·m$^{-2}$. Therefore, minimising the proportion of TS while favouring plant growth is possible, and this strategy results in much lighter GM. Moreover, this practice avoids the need to remove TS natural resource; it favours the recycling of waste produced by man for the construction of fertile GM.

The mixes with the greatest proportion of foam are most suitable when it comes to recycling PU foams. GM2 contained the greatest proportion of foam. Based on 80% of a sanitised compost:foam mixture (40:60) and a foam density of 25 kg·m$^{-3}$ on average [20], one can use about 12.5 kg of PU foam *per* m$^3$ of GM. In Germany, 8 million m$^2$ of green roofs are built every year. Assuming a GM thickness of 15 cm, we could recycle 30,000 tons of foam annually. In France, the annual production of PU waste is estimated at 250,000 tons, but only 3000 tons are recycled each year [17]. Therefore, this new recovery channel is promising.

The choice of plant species is a major issue in landscaping. In the context of this study, it was judicious to choose metal hyperaccumulators to limit losses through leaching. HY and LP had the highest metal—particularly Fe—contents, while ST had the lowest elemental content. We further calculated the stocks of elements in the plant biomass. LP stored the highest quantity of elements, even though its biomass was 2 to 5 times lower than that of EU (Tables S8 and S9). Therefore, LP is an interesting candidate species for element storage. Although HY also appeared promising, it had more difficulties growing in the GM. As green roofs can be aesthetically pleasing, the plant palette can be very diverse. A combination of hyper-accumulators and water-stress resistant species such as sedums [73] should be considered in the context of water conservation and limited irrigation.

*4.4. Study Limitations*

This study was carried out over a relatively short period (18 months), compared to the lifespan of a green roof, which is limitless as long as it is properly maintained (e.g., irrigation, fertilisation). Consequently, certain processes related to foam degradation—not quantified in the present study—may occur. Several studies conducted in soilless conditions have shown that physico-chemical degradation of foam is possible. Degradation by hydrolysis is common [74,75], and can cause pH in solution to decrease [76]. Degradation by photo-oxidation can generate secondary compounds derived from glycol [77]. Biodegradation has also been studied by incubating foams with microorganisms from compost [78] or soil [79]. PU degraders belong to the bacterial and fungal kingdoms [20,80]. In particular, mycelia can break the bonds between the polymers that make up PU [81]. However, despite some sensitivity of PU foams to biological degradation, they are not defined as biodegradable and compostable materials under the European standard EN 13432. According to this

standard, a material is considered biodegradable if degradation reaches 90% after 6 months under composting conditions, which was not the case visually in our study. Furthermore, the non-toxicity of the degradation by-products is a requirement of this standard [82].

We failed to establish a complete balance of materials because we did not quantify the initial elemental contents of the 3 GM. These initial contents were only quantified in the materials used for the mixtures (Tables 2 and 3). This balance would have enabled us to verify that the elements lost by leaching were properly quantified. In addition, we did not analyse PAH and BDE pollutants in the leachates because the initial content in the materials was below the standards. However, the risk of transfer of these elements was assumed to be low: these pollutants are complex molecules composed of aromatic rings, so that they are not very mobile. On the other hand, as they are organic molecules, they are biodegradable and may have an impact on the environment. An important scientific issue will consist in determining whether BDE debrominates in the environment into BDE congeners with fewer, maybe more toxic bromine atoms [83].

## 5. Conclusions

This study shows that the use of recycled PU foams in green roofs represents a new way of valorising this waste. From a mechanical point of view, the load of the 3 GM complies with the French and international recommendations for extensive green roofs. From an agronomic point of view, the 3 GM have physico-chemical properties favourable to plant growth. The compost:foam ratio (GM1 vs. GM2) resulted in a lower pH and higher exchangeable cations in GM1. The topsoil proportion effect (GM2 vs. GM3) affected all parameters, and particularly a lower bulk density and macroporosity in GM2, and a higher nutrient contents in GM2. Contaminant contents were all lower than recommandation thresholds, and PAH and BDE contents increased when PU foam proportion increased. GM did not affect plant biomass, and plants affected very few GM properties (i.e., exchangeable MgO and K$_2$O). However, plant affected all plant nutrient and trace metal contents. Indeed, HY and LP concentrated elements (Fe, Zn, Cu, Mn, Cr and Zn) in higher quantities than the other two species. Important losses by leaching occurred during the first 3 months, but therafter trace metal concentration and nitrogen contents were lower than the drinkable water recommendation. This work needs to be completed by ecotoxicological analyses. Furthermore, the potential biodegradation of foams and the contaminants they are likely to release—particularly PAHs and fire retardants—need to be investigated.

**Supplementary Materials:** The following are available online at https://www.mdpi.com/article/10.3390/su142013679/s1: Table S1: Contents in trace metals, volatile organic compounds (VOCs), sum of 7 polychlorinated biphenyls (PCBs), sum of 16 polycyclic aromatic hydrocarbons (PAHs) and decabromodiphenyl ethers (BDEs) of the GM at the end of the experiment. Table S2: Results of 3-way-repeated measures ANOVA with growing media (GM1 and GM2) and plants at the end of the experiment for the characteristics of the GM. Table S3: Results of 3-way-repeated measures ANOVA with growing media (GM2 and GM3) and plants at the end of the experiment for the characteristics of the GM. Table S4: Results of 3-way-repeated measures ANOVA with growing media (GM1 and GM2) and plants at the end of the experiment for the characteristics of the plant aerial parts. Table S5: Results of 3-way-repeated measures ANOVA with growing media (GM2 and GM3) and plants at the end of the experiment for the characteristics of the plant aerial parts. Table S6: Results of 3-way-repeated measures ANOVA with growing media (GM1 and GM2) and plants at the end of the experiment for the characteristics of the plant roots. Table S7: Results of 3-way-repeated measures ANOVA with growing media (GM2 and GM3) and plants at the end of the experiment for the characteristics of the plant roots. Table S8: Nutrient contents in the plant aerial biomass ($n$ = 3). Table S9: Nutrient contents in the plant root biomass ($n$ = 3). Table S10: Dynamic of the phosphate content (mg·L$^{-1}$) in the leachates. The maximum threshold contents correspond to the drinkable water standard (WHO, 2017). Table S11: Dynamic of the sulphate content (mg·L$^{-1}$) in the leachates. The maximum threshold contents correspond to the drinkable water standard (WHO, 2017). Table S12: Dynamic of the potassium content (mg·L$^{-1}$) in the leachates. The maximum threshold contents correspond to the drinkable water standard (WHO, 2017). Table S13: Dynamic of

the magnesium content (mg·L$^{-1}$) in the leachates. The maximum threshold contents correspond to the drinkable water standard (WHO, 2017). Table S14: Dynamic of the calcium content (mg·L$^{-1}$) in the leachates. The maximum threshold contents correspond to the drinkable water standard (WHO, 2017). Table S15: Dynamic of the copper content (mg·L$^{-1}$) in the leachates. The maximum threshold contents correspond to the drinkable water standard (WHO, 2017). Table S16: Dynamic of the lead content (mg·L$^{-1}$) in the leachates. The maximum threshold contents correspond to the drinkable water standard (WHO, 2017). Table S17: Dynamic of the boron content (mg·L$^{-1}$) in the leachates. The maximum threshold contents correspond to the drinkable water standard (WHO, 2017).

**Author Contributions:** Conceptualisation, P.C., M.A., O.L., H.B., L.V.-B. and R.G.; methodology, P.C., M.A., L.V.-B. and R.G.; validation, P.C., M.A., O.L., H.B., L.V.-B. and R.G.; formal analysis, P.C., M.A., L.V.-B. and R.G.; investigation, P.C., M.A., L.V.-B. and R.G.; writing—original draft preparation, P.C., M.A., L.V.-B. and R.G.; writing—review and editing, P.C., M.A., O.L., H.B., L.V.-B. and R.G.; visualisation, P.C., M.A., O.L., H.B., L.V.-B. and R.G.; supervision, P.C., L.V.-B. and R.G.; project administration, O.L.; funding acquisition, P.C., M.A., O.L., H.B., L.V.-B. and R.G. All authors have read and agreed to the published version of the manuscript.

**Funding:** This research work was funded by ECOMOBILIER, Eco-Innovation Challenge 2017, grant number 1893.

**Institutional Review Board Statement:** Not applicable.

**Informed Consent Statement:** Not applicable.

**Data Availability Statement:** Data sharing not applicable.

**Acknowledgments:** The authors are very grateful to Rémi Gardet (Phenotic platform) for the preparation of the experimental plot and for logistic support, and to Yvette Barraud-Roussel and Céline Levron (EPHor) for laboratory measurements.

**Conflicts of Interest:** The authors declare no conflict of interest. The funders had no role in the design of the study; in the collection, analyses, or interpretation of data; in the writing of the manuscript, or in the decision to publish the results.

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
