# Peer review of "Agronomic Evaluation of Recycled Polyurethane Foam-Based Growing Media for Green Roofs"

_sustainability, doi:10.3390/su142013679_

Round 1
Reviewer 1 Report
The study is about the safe use of waste from green roofs, and growing hyperaccumulators on these wastes to extract the heavy metal, then how to dispose of these heavy metal accumulators from the agri ecosystem , if dumped in the cultivated soil, these may cause further soil pollution.
Though the study is well planned and well thought out, there remains some deficit in the manuscript, as mentioned in the body of the MS. The authors have admitted to not analyzing the PAH or other secondary pollutants from GM, which if included will make the study outstanding. Still, the data and the information generated is useful and can help the academia, policy planner, etc for the safe disposable of PU

Author Response
The study is about the safe use of waste from green roofs, and growing hyperaccumulators on these wastes to extract the heavy metal, then how to dispose of these heavy metal accumulators from the agri ecosystem , if dumped in the cultivated soil, these may cause further soil pollution.
Though the study is well planned and well thought out, there remains some deficit in the manuscript, as mentioned in the body of the MS. The authors have admitted to not analyzing the PAH or other secondary pollutants from GM, which if included will make the study outstanding. Still, the data and the information generated is useful and can help the academia, policy planner, etc for the safe disposable of PU
Author thanks reviewer’s comment, we took into account all suggestions made inside the manuscript in the revised one. About your comments:
“what is the level of recycling, is this material really biodegradable”: recycled foams are understood to be foams extracted from mattress and armchair cores, "waste" could be given as a synonym. The biodegradability of these foams is questioned in the discussion and perspectives. We do not have the answer. But in the medium term (18 months experiment) this biodegradation has not been observed
“please put some data in the abstract about the salient findings” : See L18-22
L248 “This can be part of material and method, are the treatments control/regulate the prevailing climatic conditions in the study”: you’re right, we moved this part to M&M section L178
L491 “what are theses standard, please explain”
L662 “AFNOR, so many references from one source”
First we made a mistake, it was NFU44-095, we corrected. This standard precise the main characteristic a compost should have in terms of dry matter, organic matter content, C-to-N ratio trace elements and PAH limits.
“AFNOR” is the French organism that define all standard methods for soil and growing media analysis, but also plant analysis and many others
Reviewer 2 Report
This document adressess an import issue about the the use of PU foam in green roofs, which are of a great interest and are expected to achieve a higher importance in the near future. The experiments performed seem correct although some design gaps that are recognised by the authors as limitations.
Conclusions could be a little bit more developed from my humble point of view, they seem to have been written in a rush.
References seem a little bit out fashioned
Author Response
This document adressess an import issue about the the use of PU foam in green roofs, which are of a great interest and are expected to achieve a higher importance in the near future. The experiments performed seem correct although some design gaps that are recognised by the authors as limitations.
Thank you for your comment !
Conclusions could be a little bit more developed from my humble point of view, they seem to have been written in a rush.
We developed more the conclusion, L565-575
References seem a little bit out fashioned
In reality, bibliography about PU foam and particularly PU foam in substrates or soil are very scarce and some of them old. But without taking into account, the numerous analysis standard methods, 23 of the 50 references have less than 10 years old
Reviewer 3 Report
1- In Abstract author should add the main findings with quantified results
2- Line 129 : we have M1(60 :40) and M2(40 :60), in the table 1 author use M1(60 :40) and M1(40 :60), author should verify this confusion
3- Table S.6 or S4 , S7, this notation not clear ,
Author Response
In Abstract author should add the main findings with quantified results
This was done L18-22
Line 129 : we have M1(60 :40) and M2(40 :60), in the table 1 author use M1(60 :40) and M1(40 :60), author should verify this confusion
Thank you it was a mistake and we corrected
Table S.6 or S4 , S7, this notation not clear
We left the “.” In the overall manuscript and sup materials